# Production of High-Quality Wheat Sprouts of Strong Antioxidant Capacity: Process Optimization and Regulation Mechanism of Red Light Treatment

**DOI:** 10.3390/foods13172703

**Published:** 2024-08-27

**Authors:** Jing Zhang, Chunping Wang, Weiming Fang, Runqiang Yang, Yongqi Yin

**Affiliations:** 1College of Food Science and Engineering, Yangzhou University, Yangzhou 225009, China; mz120222090@stu.yzu.edu.cn (J.Z.); mz120211961@stu.yzu.edu.cn (C.W.); wmfang@yzu.edu.cn (W.F.); 2College of Food Science and Technology, Nanjing Agricultural University, Nanjing 210095, China; yangrq@njau.edu.cn

**Keywords:** red light, total phenolic, wheat sprouts, optimization, antioxidant capacity

## Abstract

Light treatment is an innovative method to enhance the synthesis of secondary metabolites in plants and improve the quality of plant-based food ingredients. This study investigated the effects of red light treatment on the physiological and biochemical changes during wheat germination, aiming to produce high-quality wheat sprouts with strong antioxidant capacity. Using response surface methodology, the study optimized the conditions for phenolic accumulation in wheat sprouts under red light treatment and explored the molecular mechanisms behind the enhancement of total phenolic content (TPC) and quality. The results indicated that red light treatment significantly increased the TPC in wheat sprouts. The highest TPC, reaching 186.61 μg GAE/sprout, was observed when wheat sprouts were exposed to red light at an intensity of 412 μmol/m²/s for 18.2 h/d over four days. Compared to no light, red light treatment significantly increased the content of photosynthetic pigments (chlorophyll and carotenoids). Red light treatment notably heightened the levels of both free and bound phenolic in the germinating wheat. Red light treatment markedly boosted the activities and relative gene expression levels of enzymes related to phenolic biosynthesis, including phenylalanine ammonia-lyase, cinnamate-4-hydroxylase, and 4-coumarate-CoA ligase. Additionally, red light treatment enhanced the antioxidant capacity of wheat sprouts by improving the activity and gene expression of four key antioxidant enzymes, thereby promoting growth and germination. This research suggested that red light treatment is an effective strategy for stimulating total phenolic biosynthesis, enhancing antioxidant capacity, and producing highly nutritious wheat sprouts, thus laying the groundwork for developing total phenolic-enriched wheat sprouts as valuable food ingredients in the future.

## 1. Introduction

Wheat (*Triticum aestivum* L.) is a widespread monocotyledonous grass plant; together with rice and maize, it stands as one of the three paramount staple food crops globally [1]. As an evolving food form, wheat sprouts are more nutritious than their raw counterparts. Certain original chemicals, including γ-aminobutyric acid, flavonoid, total phenolic, and other useful components, are highly elevated during germination [2]. As an important stimulatory metabolite in wheat, total phenolic plays an important role in the fight against cancer, the fight against bacteria and infections, and the reduction of blood sugar and cholesterol [3].

In wheat sprouts, the metabolic pathway of phenylpropane is primarily responsible for the synthesis of total phenolic [4]. The enzymatic cascade responsible for the biosynthesis of phenylpropanoids encompasses seven key enzymes, namely phenylalanine ammonia-lyase (PAL), cinnamic acid 4-hydroxylase (C4H), 4-coumaroyl-CoA ligase (4CL), p-coumarate 3-hydroxylase (C3H), caffeic acid O-methyltransferase (COMT), phenolic 5-hydroxylase (F5H), and cinnamyl alcohol dehydrogenase (CAD). The activity and gene expression of these synthetic enzymes are closely associated with total phenolic synthesis [5,6]. In addition to being controlled by developmental signals, the enzyme activity and gene expression responsible for total phenolic production in plants can also be activated by a range of external stimuli, such as light signals, temperature changes, and water stress [7,8].

Light is a significant factor in the modulation of plant growth and developmental processes., especially during germination and shoot growth, where seeds are very sensitive to changes in light [9]. Plant physiological and biochemical processes are affected by light in several ways, including the length, timing, and intensity of illumination. Plants respond differently to diverse lighting situations, which are shown mostly in morphological changes and plant growth. For example, it was discovered in the study by Smirnov et al. [10] that the weight of lettuce, both in its fresh and dry states, demonstrated an upward trend in correspondence with prolonged exposure to illumination. Azad et al. [11] noted that lettuce exposed to high levels of red light had a higher plant height and higher fresh weight. Additionally, exposure to light controls the antioxidant qualities of plants. Antioxidant enzymes and substances are important components of the plant’s antioxidant system. Studies by Nazir et al. [12] and Ahmadi et al. [13] have shown that light increases the activity of peroxidase (POD), superoxide dismutase (SOD), and ascorbate peroxidase (APX) in plants. It should be noted that a multitude of research has demonstrated that exposure to light stimulates the synthesis of useful chemicals in plants [14]. For example, it was discovered that red light improved the synthesis of phenolic in dragon heads, while blue light enhanced the concentration of total phenolic in pea sprouts [15]. Light treatment in some cereals has also garnered attention. A study examined the effects of different wavelengths of light-emitting diode (LED) radiation on free and bound phenolic compounds and antioxidant activity in einkorn and emmer sprouts and wheatgrass [16]. The results indicated specificity in light effects, showing negligible impact on sprouts but significantly greater effects on wheatgrass. Red and blue light increased the free and total content of polyphenols, tannins, flavonoids, and phenolics in sprouts. Antioxidant activity in both plants was enhanced by blue light and reduced by red light. Similarly, changes in flavonoid and anthocyanin content varied in buckwheat seedlings grown under etiolation and continuous light conditions, with an increase in flavonoid content under both treatments [17]. Rutin accumulated more under light conditions, and anthocyanin levels increased with more light exposure. Wheat sprouts emerge as a potential food source rich in bioactive compounds. It is of great value to investigate how light treatment affects the growth and biosynthesis of total phenolic in wheat sprouts.

Thus, the aims of this study encompass (1) the use of response surface methodology (RSM) to optimize the germination condition to encourage the accumulation of total phenolic content (TPC) through red light treatment and (2) investigate the effect of red light on the TPC, growth traits, and antioxidant capacity, as well as changes in the activity and relative gene expression of antioxidant enzymes and total phenolic synthase in wheat sprouts. This study can provide a scientific base and technical support for the use of red light to enhance the nutritional composition of wheat sprouts.

## 2. Materials and Methods

### 2.1. Plant Materials and Cultivation Conditions

Wheat seeds (Nongda 3753, *Triticum aestivum* L.) were supplied by Fuyichun Seed Industry Co., Ltd. (Hejian, China) in 2022 and stored at −20 °C before germination. The wheat seeds were immersed in a 1.5% (*v*/*v*) solution of sodium hypochlorite for a duration of 15 min, then rinsed repeatedly with distilled water. The washed seeds were soaked for 6 h in dark conditions at 25 °C and then spread in the germination box in the dark. Then, they were germinated for two days at 25 °C and 85% relative humidity before being put in a light incubator. Red light (660~665 nm, Shandong Guixiang photoelectric Co., LTD), relative humidity in the light incubator was 85%, temperature 25 °C and dark (no light) was used as the control (CK). Deionized water of 20 mL was sprayed every 12 h during the development period. The samples were obtained after 2 and 4 days (d) of illumination culture and stored in a conventional freezer at −20 °C until the moment of analysis.

### 2.2. Optimal Design of Germination Conditions

After conducting single-factor tests, we established the initial range for illumination intensity, duration, and period (as illustrated in Appendix A). For the Box-Behnken study design, three dependent variables were considered. The independent variables were defined with their respective lower and upper limits, as follows: illumination intensity of 300 and 500 μmol/m^2^/s, illumination time of 3 and 5 d, and illumination period of 12 and 20 h/d. The independent variables’ ranges and values are shown in Appendix A.

### 2.3. Determination of Total Phenolic Content (TPC)

The determination of the TPC, free phenolic content, and bound phenolic content was conducted following the method outlined by Ma et al. [18].

The Folin-Ciocalteu colorimetric method was employed to ascertain the phenolic content in the samples. In the experimental protocol described, a total volume of 0.2 mL of diluted crude extracts or standard solutions was mixed with 1.5 mL of freshly diluted Folin-Ciocalteu reagent at a 10-fold concentration, followed by an incubation period of 5 min to achieve equilibrium. Subsequently, 1.5 mL of sodium carbonate solution was added and thoroughly mixed. Following incubation at ambient temperature for 2 h in the absence of light, the measurement of absorbance was conducted at a wavelength of 765 nm employing a spectrophotometer. Methanol functioned as the solvent, with gallic acid utilized as the reference standard. The TPC was quantified and reported as gallic acid equivalents.

### 2.4. Determination of Physiological and Biochemical Indexes

A total of 30 wheat sprouts were randomly selected for measurement of sprout length. The methods proposed by Zhang et al. [19] and Mahdavian et al. [20] were employed to ascertain the chlorophyll and carotenoid content of wheat sprouts, respectively. The wheat sprouts were ground up using ethanol and calcium carbonate. Following this, acetone was introduced to the homogenized mixture and subsequently agitated with force. The specimen underwent centrifugation followed by the retrieval of the supernatant. The optical density was then assessed at wavelengths of 645 nm, 663 nm, and 440 nm.

### 2.5. Determination of Antioxidant Capacity

The clearance rate of 1,1-diphenyl-2-trinitrophenylhydrazine (DPPH) and 2,2-diazo-bis (3-ethyl-benzothiazole-6-sulfonic acid) diammonium salt (ABTS) was determined according to the method of Li et al. [21]. The clearance rate of ABTS was determined by reading the absorbance at 734 nm on a spectrophotometer. The determination of DPPH clearance rate was achieved through spectrophotometric analysis at a wavelength of 515 nm.

### 2.6. Determination of Antioxidant Enzyme Activity and Total Phenolic Synthetase Activity

The wheat sprouts were ground in a phosphate buffer solution (pH 7.0, 50 mmol/L), and the resulting supernatant was obtained after centrifugation. The activities of superoxide dismutase (SOD) and ascorbate peroxidase (APX) were determined following the method of Tang et al. [22]. Catalase (CAT) and peroxidase (POD) activities were tested using the methods described by Yin et al. [23]. The units of SOD and APX activity were defined as 0.01 per minute changes at OD_560nm_ and OD_290nm_, respectively. The measurement of catalase (CAT) and peroxidase (POD) activities was described as the alteration observed per minute at OD_240nm_ and O_470nm_, correspondingly, with a specific numerical value set at 0.01.

The functions of PAL, 4CL, and C4H were assessed following the methodology outlined by Ma et al. [24]. The wheat sprouts were homogenized with Tris-HCl buffer (pH 8.9, 0.1 M), and the resulting supernatant was separated from the homogenate by centrifugation. The enzymatic activity of PAL, 4CL, and C4H was standardized to one unit based on the rate of change in absorbance at 290 nm, 340 nm, and 333 nm, which was 0.01 per minute.

### 2.7. Analysis of Relative Gene Expression Levels RNA Extraction and Quantitative Real-Time PCR Analysis

The procedures of extracting RNA, conducting reverse transcription, and carrying out quantitative real-time PCR analysis of wheat sprouts were executed in compliance with the guidelines stipulated by the E.Z.N.ATM Plant RNA Kit (R6827-01, Omega, GA, USA), the PrimeScript™ RT Master Mix Kit (RR036A, Takara, Japan), and the SYBRR Premix Ex-TaqTM Kit (RR420A, Takara, Japan), respectively. The ABI 7500 sequence detection system (Applied Biosystems, Foster City, CA, USA) was used for gene expression analysis. Appendix A presents the nucleotide sequences of the primers employed in the investigation.

### 2.8. Statistical Analysis

The study was replicated thrice, with the outcomes presented as the mean accompanied by standard deviations. Tukey’s multiple range test was employed, where a statistical significance was determined by a *p*-value below 0.05.

## 3. Results

### 3.1. Optimum of Germination Conditions

The findings of the single-factor experiment indicate that the best conditions for the production of total phenolic in wheat sprouts were 400 μmol/m^2^/s of illumination intensity, 4 d of illumination time, and 18 h/d of illumination period. Based on the findings of the single-factor analysis, a Box-Behnken design comprising 17 runs was employed, incorporating three variables with three different levels each to fine-tune the parameters of light treatment and construct a secondary response surface model (Refer to Appendix A).

The statistical examination of the quadratic polynomial model and the outcomes of the analysis of variance are presented in Table 1. The adequacy of a predictive model is evaluated by calculating the coefficient of determination R-squared (R^2^), with the model in question having an R^2^ value of 0.9982. The model’s validity was confirmed through the observation of a significant F-value (*p* < 0.05), while the lack of fit F-value yielded a non-significant result (*p* > 0.05), thereby supporting its robustness in the statistical analysis. Table 1 demonstrates a notable impact (*p* < 0.01) of illumination intensity, time, and period on the TPC. Furthermore, the study revealed that the interaction variables between illumination intensity and illumination time (*p* < 0.05) and between illumination time and illumination period (*p* < 0.01) were significant for the enrichment of TPC. Substantial variances were noted among secondary variables. Equation (1) was derived through an examination of the experimental data employing the technique of multiple regression analysis.
(1)$$Y=184.92+0.8867X_{1}+1.33X_{2}+1.16X_{3}-0.5814X_{1}X_{2}+0.3963X_{1}X_{3}-0.9555X_{2}X_{3}-4.07X_{1}^{2}-11.27X_{2}^{2}-1.34X_{3}^{2}$$
Y: TPC, X_1_: illumination intensity, X_2_: illumination time, X_3_: illumination period.

As per the data illustrated in Figure 1 and the mathematical expression denoted as Equation (1), 412 μmol/m^2^/s of illumination intensity, 4 d of illumination duration, and 18.2 h/d of illumination period are the optimal lighting parameters for increasing the TPC in wheat sprouts. Under these conditions, the maximum anticipated TPC of 188.16 μg GAE/sprout was obtained.

### 3.2. Effect of Red Light on TPC

The application of red light treatment to wheat sprouts contributed to a considerable increase in the TPC (Figure 2). The TPC of the wheat sprouts was 1.26 times higher (186.61 μg GAE/sprout) than that of the control group (Figure 2I). Wheat sprouts treated with red light exhibited significantly higher amounts of bound and free phenolic compared to the control. After four days of illumination, the amount of bound and free phenolic rose by 24.20% and 28.61%, respectively, under red light compared to the CK.

### 3.3. Effects of Red Light on Physiological and Biochemical Indexes

Figure 3I,II demonstrate that the length of the wheat sprouts under red light treatment was significantly longer than that of the control group (*p* < 0.05). After four days of illumination, the length of the wheat sprouts increased to 1.50 times that of the control. As the amount of time under red light increased, the content of carotenoids and chlorophyll in wheat sprouts rose (Figure 3III,IV). The highest levels of chlorophyll and carotenoid content-27.39 and 2.81 times higher, respectively, than in the control, were attained when the illumination time reached 4 d. The clearance rates of DPPH and ATBS with the growth of wheat sprouts. Red light significantly increased the clearance rates for DPPH and ATBS of wheat sprouts (*p* < 0.05). The DPPH and ATBS clearance rates under red light were 55.49% and 36.99%, respectively, after 4 days of illumination.

### 3.4. Effects of Red Light on Antioxidant Capacity

Figure 4I,II demonstrate the clearance rates of DPPH and ATBS with the growth of wheat sprouts. Wheat sprouts under red light had much higher clearance rates for DPPH and ATBS. (*p* < 0.05). The DPPH and ATBS clearance rates under red light were 55.49% and 36.99%, respectively, after 4 days of illumination.

### 3.5. Effects of Red Light on Antioxidant Enzyme Activity and Gene Expression

The activity of POD, SOD, CAT, and APX of wheat sprouts exposed to red light increased significantly in contrast to the control group (*p* < 0.05) (Figure 5I–IV). POD, SOD, CAT, and APX activity increased by 13.71%, 35.31%, 48.45%, and 18.68%, respectively, after four days of illumination, in comparison to the control. Moreover, following four days of illumination, the expression levels of *TaPOD*, *TaSOD*, and *TaAPX* were significantly greater than the control group’s (Figure 5V,VI,VIII), with increases of 2.28, 2.60, and 2.41 times, respectively. The expression trends of *TaPOD*, *TaSOD*, and *TaAPX* were similar to their activities. After two days of illumination, the gene expression level of *TaCAT* increased by 20% compared to the control; however, after four days, there was no significant change (*p* > 0.05) (Figure 5VII). These results imply that red light stimulates antioxidant enzyme activity and relative gene expression levels.

### 3.6. Effects of Red Light on Key Enzyme Activity of Phenolic Synthesis and Gene Expression

Figure 6 illustrates how the activity of PAL, 4CL, and C4H dramatically increased as the amount of illumination time increased. (*p* < 0.05). Compared to the control, the activities of PAL, 4CL, and C4H in 4-day-old sprouts increased by 39.37%, 88.10%, and 60.96%, respectively (Figure 6I–III). The activity levels of PAL and 4CL diminished during the development of wheat sprouts in the control group, while C4H activity remained stable. Following four days of exposure to red light, the expression levels of *TaPAL*, *Ta4CL*, and *TaC4H* were 2.05, 2.21, and 2.67 times greater than the control, respectively, and these levels were in line with the trend of enzyme activity. Furthermore, under red light treatment, *TaC3H*, *TaF5H*, and *TaCAD* were markedly increased; however, *TaCOMT* did not change significantly in wheat sprouts during germination (Figure 6IV–VI).

## 4. Discussion

Phenolic is a secondary metabolite that is widely distributed in the plant kingdom and is a member of polyphenol classes [25]. Abiotic variables such as light [26], temperature [27], and moisture content [28] can induce the synthesis of phenolic in plants. Specifically, light plays a critical role in plant development and secondary metabolite production [29]. In general, the accumulation of metabolites in plants is generally greatly influenced by illumination intensity, illumination time, and illumination period [30]. In this study, wheat sprouts were treated with red light. The TPC in wheat sprouts initially increased and then decreased with increasing illumination intensity, illumination time, and illumination period. Subsequently, after using the RSM to optimize lighting, the wheat seedling with the highest TPC had 186.61 μg GAE/sprout.

Suitable illumination intensity can promote the accumulation of assimilation products and facilitate the synthesis of secondary metabolites. For example, in the study of Shin et al. [31], it was found that the contents of total flavonoids and total polyphenols in Tartary buckwheat sprouts initially increased and then decreased with the increase of illumination intensity. Similarly, with the increase in the intensity of ultraviolet light treatment, the contents of flavonoids and anthocyanins in black wheat seedlings increased initially and then decreased [32]. This aligns with the findings of the current research. Illumination time is another important factor affecting the synthesis of secondary metabolites in plants. It was found that the total polyphenol content in mung bean sprouts increased with the increase in illumination time [33]. Similarly, in the study of Van et al. [34], it was also found that increasing illumination duration promoted the synthesis of phenolic substances in sprouts. The illumination period also plays an important role in the production of plant phenols. For example, with the increase of illumination period in pea plants treated with lighting, the contents of total polyphenols and total flavonoids in pea plants also increased gradually [9]. Similarly, prolonging the illumination period is conducive to the accumulation of buckwheat phenolics [17].

Red light is recognized to exert a considerable influence on the growth of plant sprouts [23]. In terms of physiological development, red light can promote cell division and elongation, as well as stem elongation [25]. The present research also confirmed that red light significantly affects wheat seedling growth. The length of the seedling increased significantly under red light treatment (Figure 3I,II). The study by Bartucca et al. [35] and Weremczuk-Jeżyna et al. [14] also showed that red light stimulates the growth of dragonhead grass and wheat sprouts.

Plant sprouts invariably produce photosynthetic pigments, such as carotenoid and chlorophyll, when exposed to light. Carotenoids and chlorophyll are essential nutrients for human diets that have numerous health advantages [27]. Red light treatment dramatically increased the carotenoid and chlorophyll content in wheat sprouts (Figure 3III,IV). This discovery is consistent with previous studies on other plant species. Pea plants cultured under red LED had higher beta-carotene content than those cultured under other light materials [36]. This is consistent with the results of this study. Tuan et al. [37] cultured Tartary buckwheat sprouts in the environment of white light, blue light, and red light LED and found after a period of time that the total carotenoid accumulation was most abundant in the experimental group under white light irradiation. The reason for this difference is that the purpose of this study was to promote the increase of TPC in wheat sprouts rather than the enrichment of carotenoids.

In this study, the impact of red light on the antioxidant capacity of wheat sprouts was assessed by measuring the clearance rates of DPPH and ABTS. The findings illustrated in Figure 4I,II demonstrate that red light significantly improved the ability of wheat sprouts to eliminate both types of free radicals. This result confirms the beneficial effect of red light on increasing plant antioxidant capacity and is consistent with the work by Oh et al. [38] on increasing the activity of redox enzymes by red light. The activity of antioxidant enzymes and the content of antioxidants in wheat sprouts are directly correlated with their capacity to scavenge free radicals. According to the study findings, red light increases the antioxidant capacity of wheat sprouts by increasing phenolic accumulation and antioxidant enzyme activity.

Plants treated with light produce a variety of reactive oxygen species, which can damage cell structure and function, causing oxidative stress within plant cells [30]. Antioxidant enzymes can remove these oxygen-free radicals, thereby maintaining the stability of the cellular environment [39]. The results of this study showed that red light treatment had a beneficial regulatory effect on the antioxidant oxidase system of wheat seedlings, and the activities of four antioxidant enzymes (POD, SOD, CAT, and APX) were significantly increased. Notably, *TaPOD*, *TaSOD*, and *TaAPX* all showed significant increases in relative expression levels; this development is consistent with the trend in enzyme activity (Figure 5V–VIII). This strongly suggests that red light-induced the gene expression of enzymes and increased the activity of the corresponding enzymes. The activity and gene expression of antioxidant enzymes such as SOD and POD were also increased in barley seedlings grown under UV-B treatment [24]. This result confirms the important role of light signaling in regulating the antioxidant system in plants. However, the change in CAT activity did not entirely match the change in gene expression, in contrast to POD, SOD, and APX. This finding implies that changes in the initial amount of enzyme or increases in gene expression are not the only factors influencing the increase in CAT activity in wheat sprouts. Our findings, on the other hand, were more in favor of red light directly increasing CAT enzyme activity than of gene expression regulation. This illustrates the diversity and complexity of red light regulation of CAT activity, which may involve a direct effect of red light on the CAT enzyme molecules themselves or other unknown regulatory mechanisms.

It is well known that light influences secondary metabolite production in addition to plant growth and development [38]. Phenolic, an essential secondary metabolite in wheat, is a key component of the wheat seedling antioxidant system. According to this study, after red light irradiation, the TPC in wheat sprouts increased dramatically to 186.61 μg GAE/sprout. Subsequent analysis revealed that bound and free phenolic increased but that free phenolic rose far more. The findings demonstrated that the free phenolic in the wheat seedling was more susceptible to red light and that its accumulation increased higher. Phenolic production in rye sprouts is mainly through the metabolic pathway of phenylpropanoid, and the phenolic content increased in direct proportion to synthase activity [40]. The three main enzymes that start the phenolic synthesis pathway are PAL, C4H, and 4CL. Elevated activity of these enzymes can encourage the conversion of phenylalanine to trans-cinnamic acid and *p*-coumaric acid, which are building blocks for the synthesis of additional compounds of phenolic acid [41]. This study suggests that red light stimulates the formation of phenolic in wheat sprouts by increasing the activities of PAL, C4H, and 4CL, to varying degrees.

The increase in TPC was related to the change in key enzyme activity and gene expression of the phenolic synthesis pathway. The analysis of gene expression can elucidate the mechanism of enrichment of phenolic by red light treatment at the gene level. In our work, the relative gene expression of enzymes related to phenolic synthesis (PAL, 4CL, C3H, C4H, F5H, COMT, and CAD) was examined in wheat sprouts. The majority of these genes’ expression can be photoactivated [42]. Compared to the control, the relative expression of *TaPAL*, *Ta4CL*, *TaC3H*, *TaC4H*, *TaF5H*, and *TaCAD* increased significantly in wheat sprouts under red light, while the expression of *TaCOMT* remained relatively unchanged. This indicated that the effect of red light on the expression of the phenolic synthase gene is different in wheat sprouts. The increase in the relative expression levels of *Ta4CL*, *TaC3H*, *TaF5H*, and *TaCAD* indicated that the total phenolic synthesis pathway with *p*-coumaric acid as the starting material was promoted. Chen et al. [43] and Liu et al. [44] have also observed that light promotes the production of phenolic by upregulating the expression of enzyme genes associated with phenolic synthesis.

## 5. Conclusions

The optimal lighting conditions for germination were determined to be 412 μmol/m^2^/s of illumination intensity, 4 d of illumination time, and 18.2 h/d of illumination period. The chlorophyll and carotenoid contents of wheat sprouts increased significantly after red light treatment. The application of red light therapy enhanced the TPC in wheat sprouts by upregulating the gene expression of synthetases, including *TaPAL*, *Ta4CL*, *TaC4H*, *TaC3H*, *TaF5H*, and *TaCAD*, along with increasing the activity of PAL, 4CL, and C4H. Furthermore, red light treatment increased the activity of antioxidant enzymes to improve the antioxidant system of wheat sprouts.

## Figures and Tables

**Figure 1 foods-13-02703-f001:**
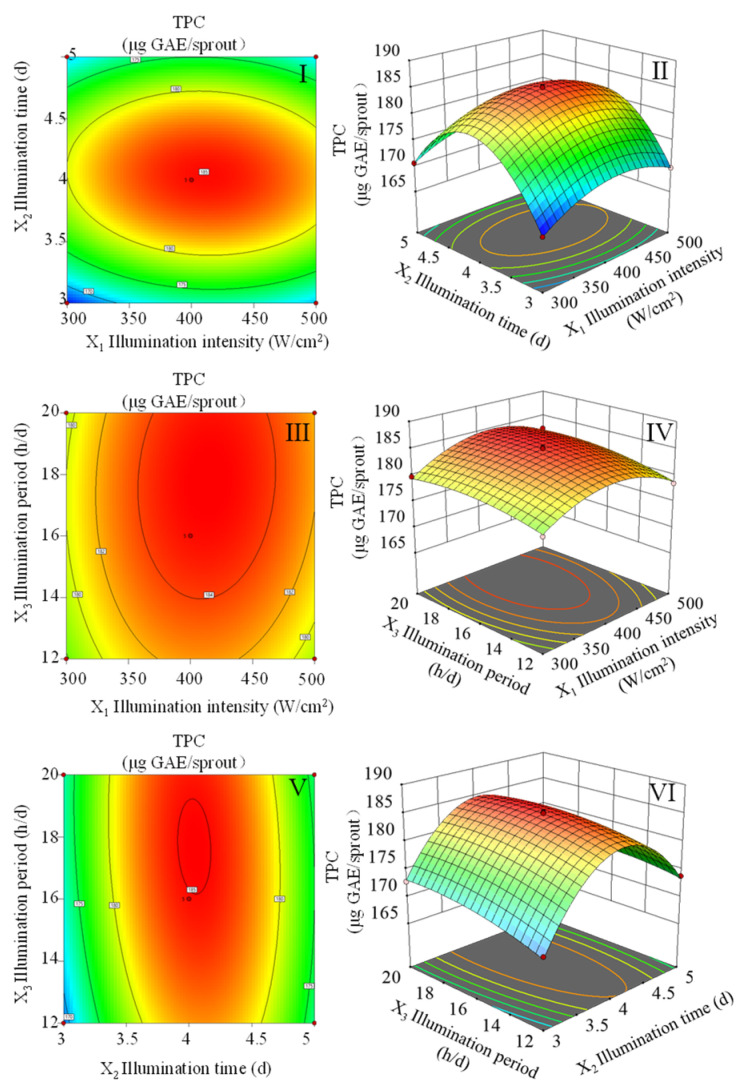
The interaction of two factors on TPC in wheat sprouts. (**I**,**II**) Illumination intensity and illumination time, and the illumination period was maintained at 16 h/d. (**III**,**IV**) Illumination intensity and illumination period, illumination time maintained at 4 d. (**V**,**VI**) Illumination time and illumination period, illumination intensity maintained at 400 μmol/m^2^/s. TPC represents the total phenolic content, and it was reported to be gallic acid equivalents.

**Figure 2 foods-13-02703-f002:**
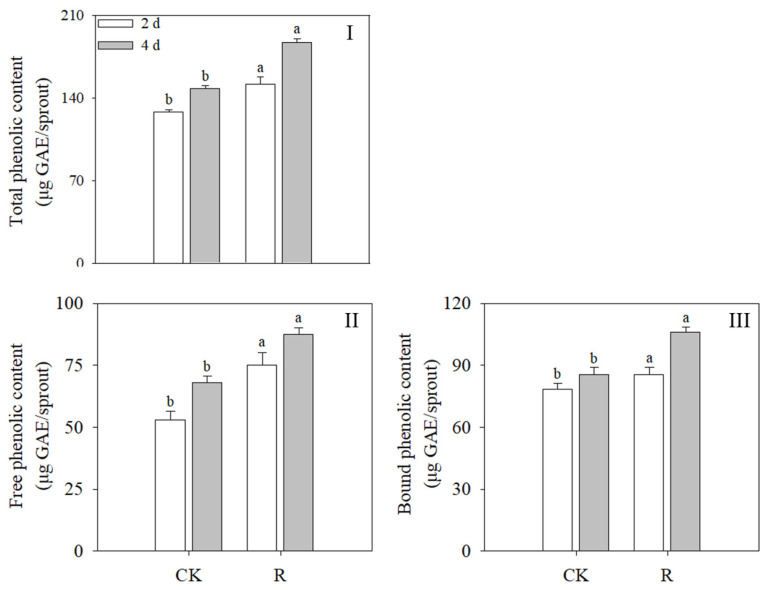
Effect of red light treatment on the content of total phenolic (**I**), free phenolic (**II**), and bound phenolic (**III**) in wheat sprouts. Distinct letters indicated the significance of variations in indices across treatments at specified germination intervals as determined by Tukey’s test (*p* < 0.05). TPC represents the total phenolic content, and it was reported to be gallic acid equivalents. CK: Control, R: Red light.

**Figure 3 foods-13-02703-f003:**
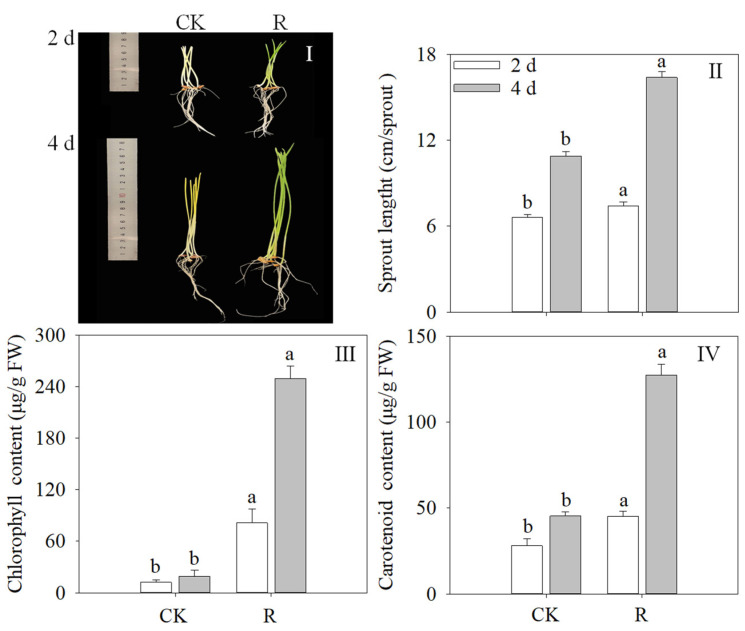
Effects of red light treatment on morphological characteristics (**I**), fresh weight (**II**), chlorophyll content (**III**), carotenoid (**IV**). Distinct lowercase letters indicated the significance of variations in indices across treatments at specified germination intervals as determined by Tukey’s test (*p* < 0.05). CK: Control, R: Red light.

**Figure 4 foods-13-02703-f004:**
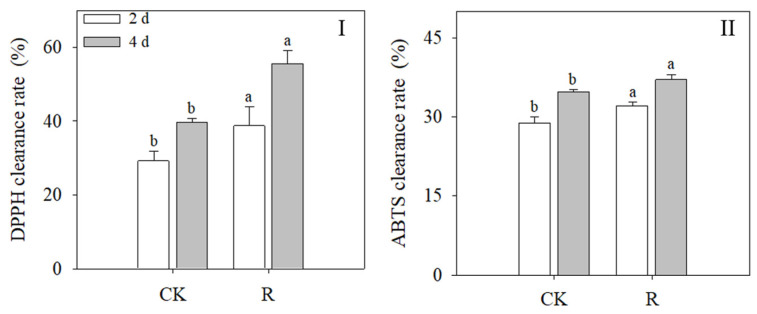
Effects of red light treatment on DPPH clearance rate (**I**) and ATBS clearance rate (**II**). Distinct letters indicated the significance of variations in indices across treatments at specified germination intervals as determined by Tukey’s test (*p* < 0.05). CK: Control, R: Red light.

**Figure 5 foods-13-02703-f005:**
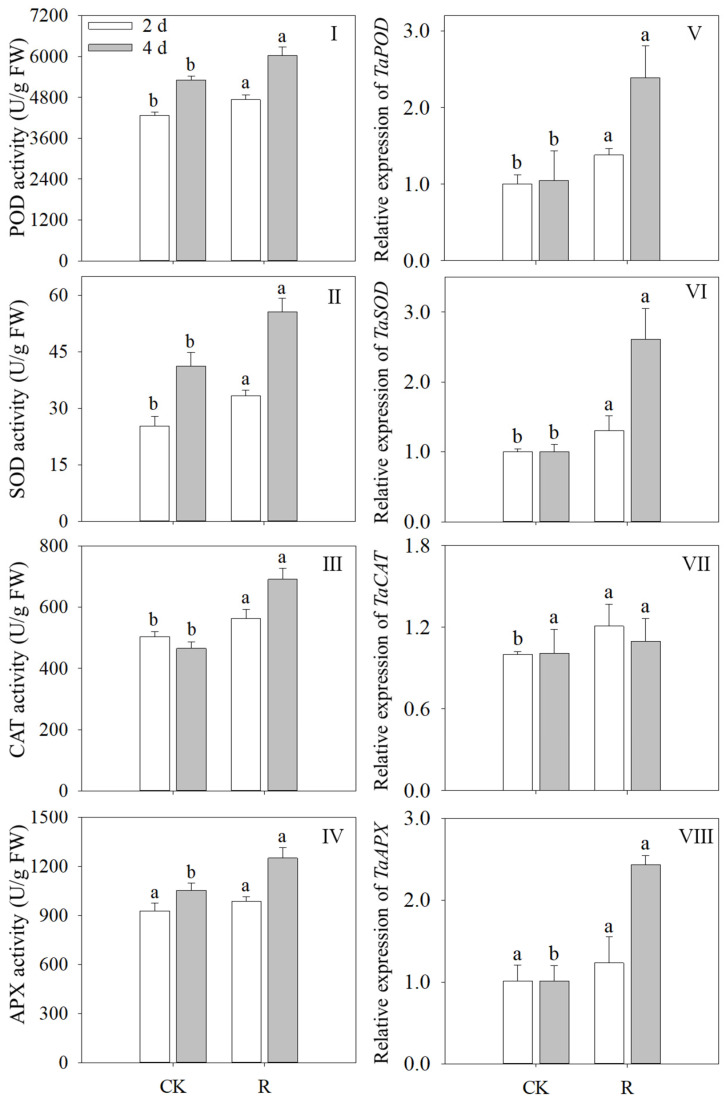
Effects of red light treatment on the activity levels of POD (**I**), SOD (**II**), CAT (**III**), APX (**IV**), and the expression level of *TaPOD* (**V**), *TaSOD* (**VI**), *TaCAT* (**VII**), *TaAPX* (**VIII**) in wheat sprouts. Distinct letters indicated the significance of variations in indices across treatments at specified germination intervals as determined by Tukey’s test (*p* < 0.05). CK: Control, R: Red light.

**Figure 6 foods-13-02703-f006:**
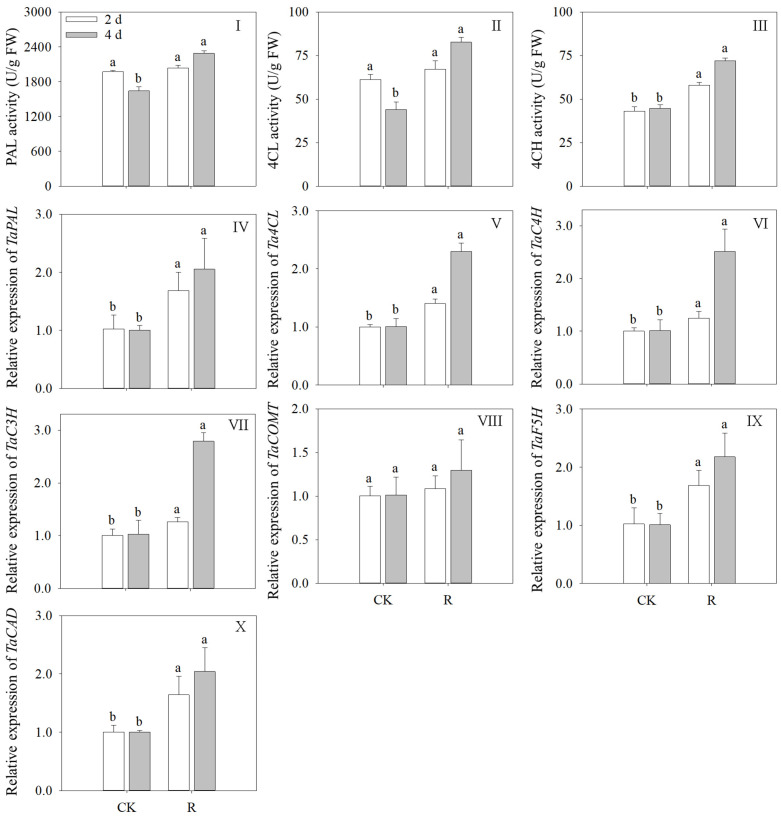
Effects of red light treatment on the activity levels of PAL (**I**), 4CL (**II**), C4H (**III**), and the expression level of *TaPAL* (**IV**), *Ta4CL* (**V**), *TaC4H* (**VI**), *TaC3H* (**VII**), *TaCOMT* (**VIII**), *TaF5H* (**IX**), *TaCAD* (**X**) in wheat sprouts. Distinct letters indicated the significance of variations in indices across treatments at specified germination intervals as determined by Tukey’s test (*p* < 0.05). CK: Control, R: Red light.

**Table 1 foods-13-02703-t001:** Analysis of the significance test for the regression coefficient and the examination of the variance equation of the regression model regression analysis.

Source	Sum of Squares	df	Mean Square	F-Value	*p*-Value
Model	684.02	9	76.00	429.10	<0.0001
$X_{1}$ Illumination intensity	6.37	1	6.37	35.98	0.0005
$X_{2}$ Illumination time	14.10	1	14.10	79.60	<0.0001
$X_{3}$ Illumination period	10.95	1	10.95	61.83	0.0001
$X_{1}X_{2}$	1.36	1	1.36	7.66	0.0278
$X_{1}X_{3}$	0.6006	1	0.6006	3.59	0.0999
$X_{2}X_{3}$	3.67	1	3.67	20.70	0.0026
$X_{1}^{2}$	70.13	1	70.13	399.33	<0.0001
$X_{2}^{2}$	534.44	1	534.44	3017.34	<0.0001
$X_{3}^{2}$	7.52	1	7.52	42.45	0.0003
Residual	1.24	7	0.1771		
Lack of fit	1.03	3	0.3421	6.41	0.0522
Pure error	0.2134	4	0.0533		
Total	685.26	16			
R^2^ = 0.9982		Adjusted R^2^ = 0.9959		Predicted R^2^ = 0.9755	
Coefficient of variation = 0.2377%		Adeq precision = 56.2997			

## Data Availability

The original contributions presented in the study are included in the article/Appendix A, further inquiries can be directed to the corresponding author.

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
