# Peer review of "Production of High-Quality Wheat Sprouts of Strong Antioxidant Capacity: Process Optimization and Regulation Mechanism of Red Light Treatment"

_foods, 2024, doi:10.3390/foods13172703_

Round 1

Reviewer 1 Report

Comments and Suggestions for Authors

The paper describes the impact of red light treatment on the antioxidant activity of wheat sprouts. The idea is interesting and aligns with the general trend of enhancing food products. However, I have several comments that need to be addressed by the authors:

Main concerns:

  • It is unclear what was considered the control—darkness (line 89) or white light (line 21). The values for radiation intensity are missing in materials and metods.
  • The term used by the authors, "phenolic acid content," is incorrect because the authors examined antioxidant capacity expressed as TPC (https://doi.org/10.1021/acs.jafc.3c04022).
  • The methodology is written too briefly and should be expanded.
  • The references provided in the paper are incorrect (see detailed comments).

·        In the title of the paper and abstract, I would suggest changing "rich in phenolic acids" to "with high total phenolic content value" (or "with high antioxidant capacity"). Similarly, this adjustment should be made throughout the entire paper. It is difficult to determine whether the enrichment is with polyphenolic acids or with other compounds capable of reacting with FC reagents. The TPC test is based on an electron-transfer reaction where the antioxidant species is the electron donor and the FC reagent acts as the oxidant. This is not a quantitative method but rather is considered a test of antioxidant capacity.

Minor issues:

·  Line 20: Incorrect unit—typically expressed as gallic acid equivalent.

·  I would suggest removing or replacing "phenolic acid" with "TPC" or "antioxidant activity."

·  Throughout the paper, instead of "kwas fenolowy," the term "phenolic acids" should be used.

·  Line 41: Incorrect references selected, e.g., the cited work does not pertain to the biological properties of phenolic acid.

·  In many places in the paper, e.g., the citation method is incorrect: it should be "Azad et al." on line 57. Additionally, there are incorrect reference numbers in the cited sections on lines 105, 111, 109, 116, 247, etc. Carefully check the entire paper.

·  Unexplained abbreviations on line 63.

·  Line 70-71: The sentence should be removed.

·  The language should be revised, e.g., the passive voice is typically used in scientific texts, e.g., "was sprayed" on line 90.

·  Lines 104-105: Lack of details regarding the analyses performed; references are incorrectly cited.

·  Too few details in the methodology regarding extraction; how were the extracts containing free and bound phenolic acids obtained?

·  Paragraph 2.4: DPPH and ABTS are antioxidant tests—they should be in a separate section rather than under "biochemical indexes."

·  In Figure 1, the unit is incorrect, as is the term "contente."

·  Lines 171-172: These are not quantitative data, as they are still TPC values. TPC values obtained after hydrolysis...?

Author Response

Thank you for your valuable comments and suggestions on our manuscript (ID: foods- 3165953) ' Production of high-quality wheat sprouts rich in phenolic acids: Process optimization and regulation mechanism of red light treatment '. We have carefully revised the manuscript and resubmitted it to you for further review. (All changes can be seen in the latest manuscripts).

Response to Reviewer 1

Comments 1

The paper describes the impact of red light treatment on the antioxidant activity of wheat sprouts. The idea is interesting and aligns with the general trend of enhancing food products.

Response:

We deeply appreciate your positive evaluation of our research and manuscript, thank you!

Comments 2

It is unclear what was considered the control—darkness (line 89) or white light (line 21).

Response:

Thank you for your constructive comments The control group in this study was darkness. And we have modified the corresponding content in the manuscript.

Comments 3

The values for radiation intensity are missing in materials and methods.

Response:

Thank you for your insightful comments. In this manuscript, we adjusted the light intensity to achieve a controlled range (300-500 μmol/m2/s) by varying the distance between the sprouts and the light source using a photoradiometer (HD 2302.0, Delta OHMSRL, Italy Marconi). The unit of light intensity, μmol/m²/s, is commonly used in studies dealing with LED lighting treatments for plant seedlings.

Comments 4

The term used by the authors, “phenolic acid content,” is incorrect because the authors examined antioxidant capacity expressed as TPC (https://doi.org/10.1021/acs.jafc.3c04022).

Response:

Revised. Thank you.

Comments 5

The methodology is written too briefly and should be expanded.

Response:

Thanks. The method part has expanded in the revised manuscript.

Comments 6

The references provided in the paper are incorrect (see detailed comments).

Response:

Thank you for pointing out. We have carefully checked and revised the cited references.

Comments 7

In the title of the paper and abstract, I would suggest changing "rich in phenolic acids" to "with high total phenolic content value" (or "with high antioxidant capacity"). Similarly, this adjustment should be made throughout the entire paper. It is difficult to determine whether the enrichment is with polyphenolic acids or with other compounds capable of reacting with FC reagents. The TPC test is based on an electron-transfer reaction where the antioxidant species is the electron donor and the FC reagent acts as the oxidant. This is not a quantitative method but rather is considered a test of antioxidant capacity.

Response:

Thank you for pointing out this issue. As per the suggestions "rich in phenolic acids" in the title and abstract has been changed to "strong antioxidant capacity".

Comments 8

Line 20: Incorrect unit—typically expressed as gallic acid equivalent.

Response:

Agreed, we have adjusted the units to gallic acid equivalent in the revised manuscript.

Comments 9

I would suggest removing or replacing "phenolic acid" with "TPC" or "antioxidant activity."

Response:

Thank you for your comments. As per the suggestions we have replaced “phenolic acids” with “TPC”.

Comments 10

Throughout the paper, instead of "kwas fenolowy," the term "phenolic acids" should be used.

Response:

Thank you for your comments. It has been revised and there is no “kwas fenolowy” in the revised manuscript.

Comments 11

Line 41: Incorrect references selected, e.g., the cited work does not pertain to the biological properties of phenolic acid.

Response:

Thank you for your careful comments. We have made revisions and replaced the correct references.

Comments 12

In many places in the paper, e.g., the citation method is incorrect: it should be "Azad et al." on line 57. Additionally, there are incorrect reference numbers in the cited sections on lines 105, 111, 109, 116, 247, etc. Carefully check the entire paper.

Response:

Thank you for your serious comments. We have carefully examined the references throughout the manuscript. The reference format and reference number have been revised, and the correct citations have been re-made.

Comments 13

Unexplained abbreviations on line 63.

Response:

Thank you. We have revised it and added the full name of the abbreviation on page 2, lines 66-67.

Comments 14

Line 70-71: The sentence should be removed.

Response:

Removed.

Comments 15

The language should be revised, e.g., the passive voice is typically used in scientific texts, e.g., "was sprayed" on line 90.

Response:

Agreed. The language has been modified in the revised manuscript.

Comments 16

Lines 104-105: Lack of details regarding the analyses performed; references are incorrectly cited. Too few details in the methodology regarding extraction; how were the extracts containing free and bound phenolic acids obtained?

Response:

The authors are very sorry for the error due to our carelessness. We have revised the method. The methods of phenolic extraction and determination have been supplemented and the correct references have been replaced.

The determination of total phenols, free phenols, and bound phenols was conducted following the method outlined by Ma et al. ( https://doi.org/10.1016/j.foodchem.2018.07.092).

Free phenolic compounds: The subtle milled sample (1 g) was extracted with 80% methanol (10 mL) and shaken with a laboratory rotary shaker under nitrogen and dark conditions at 200g and 25 °C for 1 h. The mixture was then centrifuged at 10,000 g for 10 min at 4 °C. After centrifugation, the methanol supernatants were filtered and extraction was repeated three times. Then, the total supernatants was evaporated to dryness under vacuum at 40 °C with a rotary vacuum evaporator. The dried extracts were redissolved with 50% methanol to a volume of 10 mL and used as crude free phenolic extracts.

Bound phenolic compounds: The residue after ethanol extraction was mixed with 20 mL of 2 M NaOH at 25 °C for 4 h. The hydrolyzed mixture was adjusted to a pH of 1.6­-1.8 with 6 M HCl and extracted three times with 25 mL ethyl acetate. Then the mixtures were centrifugated and the combined ethyl acetate fractions were evaporated to dryness. The dried extracts were redissolved in 50% methanol (10 mL) and used as crude bound phenolic extracts. The extracts were stored at −20 °C for subsequent use.

The total phenolic content was determined using the Folin-Ciocalteu colorimetric method. Which simply means, 0.2 mL of appropriately diluted crude extracts or standard solutions were blended with 1.5 mL of 10-fold freshly diluted Folin-Ciocalteu reagent and equilibrate for 5 min, then 1.5 mL sodium carbonate solution (75 g/L) was added and mixed completely, after storing at room temperature for 2 h keeping away from light, the absorbance was measured at 765 nm using a spectrophotometer. Methanol was used as the blank, and gallic acid (GA) was used as the standards. The total phenolic content was expressed as gallic acid equivalents.

Comments 17

Paragraph 2.4: DPPH and ABTS are antioxidant tests—they should be in a separate section rather than under "biochemical indexes."

Response:

Agreed. As per the suggestions we have split the DPPH and ABTS in Figure 3 to form Figure 4.

Comments 18

In Figure 1, the unit is incorrect, as is the term "contente."

Response:

Thank you for your comments. We have revised Figure 1. The correct units were changed and the wrong words were corrected.

Comments 19

Lines 171-172: These are not quantitative data, as they are still TPC values. TPC values obtained after hydrolysis...?

Response:

Thanks for your careful comments. As indicated in comment 16, we separately measured the contents of free and bound phenolic compounds following the procedures outlined in the references.

Reviewer 2 Report

Comments and Suggestions for Authors

Dear Authors,

Thank you for submitting your manuscript to Foods. I have reviewed the manuscript, and I have attached my report. Please, re-arrange the discussion section following the pattern of results section, i.e. 3.1-3.5. Currently, the discussion does not flow and needs major revision.

Comments on the Quality of English Language

Minor editing will be required

Author Response

Thank you for your valuable comments and suggestions on our manuscript (ID: foods- 3165953) ' Production of high-quality wheat sprouts rich in phenolic acids: Process optimization and regulation mechanism of red light treatment '. We have carefully revised the manuscript and resubmitted it to you for further review. (All changes can be seen in the latest manuscripts).

Response to Reviewer 2

Comments 1

I would like to appreciate author's efforts to submit the manuscript entitled: Production of high-quality wheat sprouts rich in phenolic acids: Process optimization and regulation mechanism of red light  treatment. Major corrections will be required, below are my comments and suggestions to further improve the quality of the manuscript.

Response:

Thank you for spending time in reviewing our manuscript and providing us with a list of constructive comments.

Comments 2

The abstract, what about the effect of red light treatment on the chlorophyll and carotenoid content?

Response:

Thank you for pointing out. We have added to the abstract the effects of red light on chlorophyll and carotenoid content on lines 21 and 22 of page 1.

Comments 3

Introduction, line 63, authors must write full names of abbreviations.

Response:

Agreed. We have written the full abbreviated name in the revised manuscript on lines 66 and 67 on page 2.

Comments 4

The introduction is well written; however, I wonder if there are no related study done on another cereal seedling. This is because authors focused more on the effect of light on lettuce rather than cereal seedling. The results of lettuce and wheat seedling will have big variations since they are not related (lettuce – vegetable and wheat seedling – cereal). Thus, there is a need for authors to research deeper on the effect of light treatment on cereal seedling. This will make readers to understand the knowledge gap, for now the gap is missing.

Response:

Thank you. We have updated the relevant research on light treatment in grain seedlings (wheatgrass, and buckwheat seedlings) in the introduction.

Comments 5

Materials and methods. Line 86, should be, “Then they were germinated…”rephrase line 90 “Spray 20 mL of deionized water…” write the sentence in past tense. Same is applicable for lines 104-105.

Response:

Agreed. As per the suggestions the sentence in line 86 has been revised to read "Then they were germinated...". In addition, two grammatical errors in the sentence have been corrected in the past tense.

Comments 6

Are Che, Ma, Li, Wang and Tang, Xu in lines 105, 111 all surnames?

Response:

We appreciate your valuable feedback. The author made errors in the citation format, which we have corrected. Additionally, we have reviewed and revised the subsequent citation formats to ensure accuracy.

Comments 7

Results. Lines 131, 166, 179, 195 and 212, change to 3.1-3.5.

Response:

Agreed. As per the suggestions the subtitle serial number has been modified to the correct format in the revised manuscript.

Comments 8

Line 141, check “wa” between F-value and table significant.

Response:

Thank you for pointing out. We have checked and corrected the error.

Comments 9

Figure 2, authors must show the germination days as done in Figure 3. Currently, the bar charts are meaningless.

Response:

Thank you for pointing out. As per the suggestions we have added a legend of the number of germination days in Figure 2.

Comments 10

Line 181, should be, “was significantly higher…”

Response:

The authors are sorry for our careless mistakes. As per the suggestions we have changed “is” to “was”.

Comments 11

Lines186-187, “The clearance rates of DPPH and ATBS with..” revise because it is incomplete.

Response:

Revised.

Comments 12

Discussion. The discussion is too general, for example, lines 231-236, authors are just narrating the story about phenolic acid including various factors affecting its synthesis. Discussion must follow the pattern of sections (3.1 to 3.5) under results. In these lines (231-241), authors must explain why wheat seedlings treated with red light exhibited significantly higher amounts of bound and free phenolic acid after 4 days of illumination compared to the control.

Response:

Thank you for your serious comments. We have revised. The order has been adjusted following the pattern of the results section (3.1-3.5) as suggested. Red light treatment promoted the accumulation of phenolic acids in wheat sprouts by upregulating the gene expression of phenolic acid biosynthetic enzymes TaPAL, Ta4CL, TaC4H, TaC3H, TaF5H, and TaCAD, as well as increasing the activities of PAL, 4CL, and C4H. This point is set out in paragraphs 4 and 5 of the discussion. We have added a section discussing the impact of three individual factors under light treatment on the synthesis of bioactive substances in different plants (black wheat, buckwheat, and mung bean) in paragraph 2 of the discussion.

Comments 13

Section 3.3 of results. Lines 242-244 are too general while lines 244-246 are repetition of section 3 (results). Same trend of narrating the story is seen, for examples, lines 251-253 are too general, lines 253-254 is a repetition of results section, I do not see the relevance of lines 248-250 since basil and  tomato are vegetables while wheat is a cereal. Comparison of study must be of the same species (cereal-cereal). In this section, authors must explain why the length of sprout was highest after 4 days of illumination, same is applicable of chlorophyll, carotenoid content, DPPH and ATBS.

Response:

Thank you for your insightful comments. We have revised the general and repetitive sections you pointed out and removed irrelevant discussions.

Additionally, in our discussion of changes in chlorophyll and carotenoids, we have excluded non-grain studies and included new studies on wheat and tartary buckwheat for comparison. In this study, we found that red light treatment effectively promoted the biosynthesis of phenolic compounds in wheat sprouts. Through response surface methodology, we determined that the total phenolic content was highest on the fourth day of germination under red light treatment. We further investigated the physiological and biochemical changes in sprouts germinated for four days. The results showed a significant increase in the activity of antioxidant enzymes and the relative expression levels of their genes in the sprouts. We propose that red light treatment activates the antioxidant system in the sprouts, thereby promoting their growth and enhancing chlorophyll, DPPH, and ATBS contents.

Comments 14

Section 3.4 of results, lines 263-270 are too general and are not related to the results of the study. Lines 271-273 is the repetition of results section. The comparison using vegetables in lines 273-274  is not proper. In this section, authors must explain why POD, SOD, CAT, and APX activity  increased after four days of illumination, in comparison to the control.

Response:

Thank you for your comments. We have revised the generalized and repetitive content in the discussion section. We have removed the examples pertaining to vegetables and instead, used examples related to wheat studies (https://doi.org/10.1016/j.foodchem.2018.09.158) for comparison.

Our thoughts on the changes in antioxidant enzymes are as follows: In this study, red light served as an exogenous factor that regulated the physiological responses in the growth process of wheat seedlings, including the antioxidant enzyme system. Our research aimed to systematically understand the physiological and biochemical changes under treatment, observing the activity changes in four types of antioxidant enzymes under light treatment. How these changes occur is unpredictable—they may be stress-induced or promotive, but they are ultimately adaptive responses to external environmental changes. The results of this study showed an increase in the activity of the four antioxidant enzymes under red light treatment. This phenomenon has also been observed under treatments with salicylic acid (doi: 10.1002/jsfa.13365) and GABA (doi: 10.1016/j.foodchem.2018.07.092). Thanks again for your comments。

Comments 15

Section 3.5. authors must explain why (i) the activities of PAL, 215 4CL, and C4H in 4-day-old  seedlings increased (ii) after two days of illumination, the gene expression level of TaCAT  increased by 20% compared to the control. (iii) Following four days of exposure to red light, the  expression levels of TaPAL, Ta4CL, and TaC4H were greater than the control. (iv) under red light  treatment, TaC3H, TaF5H, 221 and TaCAD were markedly increased.

Response:

Thank you for your careful review and valuable comments.

PAL, C4H, and 4CL are three key enzymes in the phenolic acid biosynthesis pathway, with enhanced enzymatic activities promoting the synthesis of substances like phenolic acids. TaPAL, Ta4CL, and TaC4H are genes encoding the critical enzymes for phenolic compounds, while TaC3H, TaF5H, and TaCAD encode enzymes downstream in the phenolic acid biosynthesis pathway. The correlation between gene expression and enzyme activity is consistent in some studies but not in others. This study aims to enrich phenolic acids, a secondary metabolite, in wheat seedlings. For wheat, we selected light treatment, and through preliminary screening, we identified red light as a beneficial exogenous treatment, subsequently optimizing the illumination conditions. We then explored why red light enriches phenolic acids. Our findings suggest that the increase in phenolic acids may be due to the enhanced activity and expression of these key enzymes involved in phenolic acid synthesis.

Comments 16

Lines 288-290 are too general, while lines 290-293 are repetition of results section. Lines 295-297 must be linked with the results of the study, not just to mention phenolic acid production in rye seedling. Line 304, where is that strong correlation authors talk about? Lines 308-312 is the repetition of results section. Lines 318-319 is a speculation since ferulic acid was not analysed. Line 321, why cucumber seedling? Lines 321-325, authors must learn to avoid speculation when  discussing results. Line 327, should be, antioxidant capacity not properties. Lines 327-326 makes sense because they align with the results of the study. Lines 337-338 is repetition of results section. Lines 339-346 is repetition of some of the lines of the discussion section.

Response:

Thank you for your thoughtful comments. We have revised the sections identified as vague and repetitive, removing inaccurate statements such as "strong correlation." Additionally, we have edited out incorrect speculations and eliminated the poor comparative example of "cucumber." These revisions pertain to the last paragraph of the discussion.

Comments 16

Lines 337-338 is repetition of results section. Lines 339-346 is repetition of some of the lines of the discussion section. Line 339, did author measured photosynthesis to come to the conclusion? Otherwise, it is a speculation.

Response:

Thank you for your careful and earnest comments. This paragraph speculates on the mechanism of phenolic acid synthesis in wheat malt under full red-light treatment. After careful consideration, the author has decided to delete this section from the discussion, as its removal will not affect the manuscript's content.

Comments 17

Conclusion- what about the effect of red light on the chlorophyll, carotenoid?

Response:

Thanks for your valuable comment. We have added to the conclusions about the effects of red light on chlorophyll and carotenoid content in wheat sprouts.

Comments 18

References – check whether the referencing style of Foods uses DOI

Response:

Thank you for your comments. We have carefully checked the reference style and modified it according to the FOODS reference style.
